# A cryogenic, coincident fluorescence, electron, and ion beam microscope

**Daan B Boltje[1,2]\*, Jacob P Hoogenboom[1]\*, Arjen J Jakobi[3], Grant J Jensen[4,5], Caspar TH Jonker[2], Max J Kaag[1], Abraham J Koster[6], Mart GF Last[2,6], Cecilia de Agrela Pinto[3], Jürgen M Plitzko[7], Stefan Raunser[8], Sebastian Tacke[8], Zhexin Wang[8], Ernest B van der Wee[1], Roger Wepf[9], Sander den Hoedt[2]**

[1]Department of Imaging Physic, Delft University of Technology, Delft, Netherlands; [2]Delmic B.V, Delft, Netherlands; [3]Kavli Institute of Nanoscience, Delft University of Technology, Delft, Netherlands; [4]California Institute of Technology, Pasadena, United States; [5]Brigham Young University, Provo, United States; [6]Department of Cell and Chemical Biology, Leiden University Medical Center, Leiden, Netherlands; [7]CryoEM Technology, Max Planck Institute of Biochemistry, Martinsried, Germany; [8]Department of Structural Biochemistry, Max Planck Institute of Molecular Physiology, Dortmund, Germany; [9]Centre for Microscopy and Microanalysis, University of Queensland, Brisbane, Australia

\*For correspondence:
boltje@delmic.com (DBB);
J.P.Hoogenboom@TUDelft.nl
(JPH)

**Abstract** Cryogenic electron tomography (cryo-ET) combined with subtomogram averaging, allows in situ visualization and structure determination of macromolecular complexes at subnanometre resolution. Cryogenic focused ion beam (cryo-FIB) micromachining is used to prepare a thin lamella-shaped sample out of a frozen-hydrated cell for cryo-ET imaging, but standard cryo-FIB fabrication is blind to the precise location of the structure or proteins of interest. Fluorescence-guided focused ion beam (FIB) milling at target locations requires multiple sample transfers prone to contamination, and relocation and registration accuracy is often insufficient for 3D targeting. Here, we present in situ fluorescence microscopy-guided FIB fabrication of a frozen-hydrated lamella to address this problem: we built a coincident three-beam cryogenic correlative microscope by retrofitting a compact cryogenic microcooler, custom positioning stage, and an inverted wide-field fluorescence microscope (FM) on an existing FIB scanning electron microscope. We show FM controlled targeting at every milling step in the lamella fabrication process, validated with transmission electron microscope tomogram reconstructions of the target regions. The ability to check the lamella during and after the milling process results in a higher success rate in the fabrication process and will increase the throughput of fabrication for lamellae suitable for high-resolution imaging.

## Editor's evaluation

This paper is of particular interest to researchers who plan to use focused-ion beam scanning electron microscopes (FIB-SEMs) and require fluorescent data to guide the milling process. The authors describe a valuable after-market upgrade that allows fluorescent data acquisition during FIB-milling without stage repositioning. Technical details of the fluorescent module upgrade together with the sample stage redesign are compellingly documented and will enhance the implementation of this important technology.

## Introduction

High-resolution 3D reconstructions of biological macromolecules in their near-native cellular environment are necessary to obtain a mechanistic understanding of complex, biological processes at the molecular scale. Cryogenic electron tomography (Cryo-ET) allows imaging cellular structures at unprecedented resolution and clarity (*Zimmerli et al., 2021*; *Tegunov et al., 2021*). In cryo-ET, a sample is flash cooled, thinned to the appropriate thickness, and a tilt series of projections is acquired using a transmission electron microscope (TEM) (*Koning et al., 2018*; *Turk and Baumeister, 2020*). A prerequisite for high-resolution cryo-ET is that the sample is thinner than the inelastic mean-free path of electrons in vitreous water ice—in practice, this means a sample thickness of approximately 100 to 200 nm (*Vulović et al., 2013*). To create sufficiently thin sections (lamellae) for high-resolution tomography, the use of a focused ion-beam scanning electron microscope (FIB-SEM) has proven to be most successful in ablating the excess cellular material surrounding the region of interest (ROI) (*Chiang et al., 2007*; *Villa et al., 2013*; *Hylton and Swulius, 2021*). In recent years, various improvements and refinements have been made to the cryogenic focused ion beam (cryo-FIB) milling workflow improving throughput, reliability, sample yield, and quality (*Schaffer et al., 2019*; *Wolff et al., 2019*; *Tacke et al., 2021*; *Buckley et al., 2020*).

Identification of the ROI for milling in the right location is a crucial step. Unfortunately, neither the SEM nor the FIB provides a contrast mechanism for biomolecular composition, leading to blind milling and possibly inadvertent ablation of the structure of interest. Cryogenic correlative light and electron microscopy (cryo-CLEM) can be employed to overcome this challenge. In this approach, the location of specific objects or cellular compartments is targeted for cryo-FIB milling using specific fluorescent labeling, thus allowing targeted FIB milling.

The first use of cryo-CLEM for fluorescence targeted FIB milling employed monosized ferromagnetic polystyrene beads as fiducial markers for fluorescence microscope (FM) and FIB-SEM correlation (*Arnold et al., 2016*). The signal from the iron oxide in the magnetic beads is detectable in the FIB-SEM by the back-scattered electron detector and the polystyrene is autofluorescent when excited by green light in the FM. FM imaging for target localization was done in a stand-alone cryogenic spinning disk confocal microscope after which the sample was transferred to the FIB-SEM. Correlation was achieved by 3D coordinate transformation and a correlation accuracy on the order of tens of nanometer was shown. However, the error range remained relatively large, ranging from 0.2 to 1 µm depending on the number of fiducial markers used in the correlation. Arnold et al. estimated a success rate of 60% when milling site-specific lamellas of 200 to 300 nm thick which would drop when aiming for a thickness of 100 to 200 nm.

More recently, *Gorelick et al., 2019* equipped an FIB-SEM with an in situ FM, simplifying sample handling and reducing the risk of sample contamination by limiting the number of cryo-transfer steps. Switching between FM and FIB-SEM imaging modalities required repositioning of the sample inside the vacuum chamber. The 2D positional error made in milling based on FM data is mostly determined by the relocation precision of the sample stage and was reported to be in the range of <420 nm along $X$ and <220 nm along $Y$. These relatively large relocation errors prevented high enough localization accuracy to mill a lamella required for high-resolution tomograms, in a targeted fashion.

Coincident imaging with all three microscopes could mitigate the need for fiducial markers and the occurrence of relocation errors, thereby facilitating the application of cryo-CLEM for the cryo-ET workflow. Such a geometry would also allow FM imaging while milling, allowing real-time correction of the milling process in the case of errors, drift, or other misalignments.

Here, we present a coincident three-beam cryo-CLEM solution, by mounting a tilted light objective lens (OL) inside the vacuum chamber such that widefield FM imaging can be done whilst FIB milling. The light microscope (LM) is combined with a compact cryogenic microcooler and a custom positioning stage. Our system can be integrated with existing dual beam systems to allow simultaneous, coincident imaging with both FM, SEM, and FIB for in situ FM-guided fabrication of frozen-hydrated lamellae.

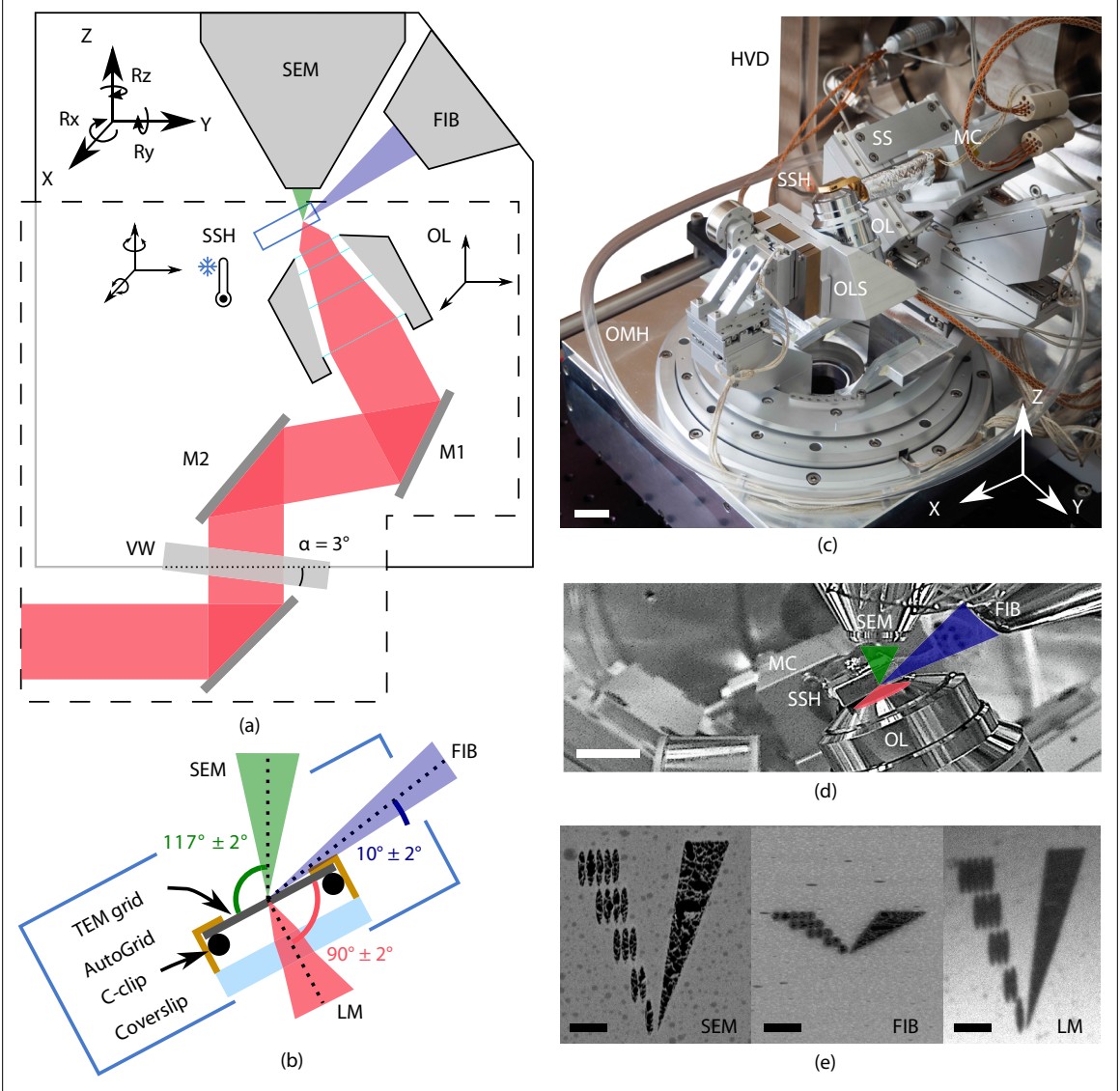

**Figure 1.** Cryogenic, coincident fluorescence, electron, and ion beam microscope. (**a**) Schematic illustration showing the retrofitted hardware (dashed box) separated from the focused ion-beam scanning electron microscope (FIB-SEM) hardware in, respectively, the lower and upper half-spaces of the microscope. (**b**) Schematic illustration of the sample shuttle holder (SSH) with the transmission electron microscope (TEM) grid (sample) clipped into an AutoGrid placed on top of the coverslip. The coincident point is formed by the electron-, ion-, and light beams. Incident angles indicated are limited by the FIB access hole in the sample shuttle holder. (**c**) Photograph of the setup showing the HV door (HVD), optical module housing (OMH) on which the sample positioning stage (SS), objective lens stage (OLS), objective lens (OL), and microcooler (MC) are mounted. Scale bar 2 cm. (**d**) Infrared photograph acquired with the charge-coupled device (CCD) camera mounted on the FIB-SEM host system showing the microcooler and OL mounted inside the HV chamber. Scale bar 10 mm. (**e**) Images of an alignment pattern in the three imaging modalities, the light microscopy image is acquired by collecting reflected light from the sample. The alignment pattern is milled in the TEM grid coating using the FIB. Scale bar 3.5 μm.

## Results
### System description
#### System overview

A principal design element of our microscope is the placement of the OL of the FM directly below the sample (*Zonnevylle et al., 2013*). This minimizes interference with the electron microscope (EM) hardware residing in the upper half space of the EM and offers the following advantages (*Figure 1*): (1) the OL focal point can be aligned to the FIB-SEM working point, allowing simultaneous and coincident imaging with electron-, ion-, and photon beams. (2) In situ FM-guided FIB milling of frozen-hydrated lamella without any stage moves can be conducted. (3) Quality control of the lamella fabrication

process can be performed with the LM at every step in the milling process. (4) Multiple ROIs on the sample can be processed, by centring and focusing the fluorescent ROI in the LM image after a single, initial OL alignment. (5) An OL with a relatively high numerical aperture (NA) can be used, in our case an NA of 0.85, allowing high-resolution fluorescence localization.

Vitrified cellular samples on TEM AutoGrids (*Rigort et al., 2012*) are positioned in between the OLs of all three microscopes. The AutoGrids are located on top of a thin indium tin oxide (ITO)-coated coverglass that serves as a beam stop to prevent exposure of the OL to electrons or ions. A cryogenic microcooler is used to keep the sample vitrified without the need to cool any of the microscope components, thus retaining unmodified imaging modalities of all three beams. The FM OL and microcooler can be moved about in the vacuum by custom piezo positioning stages. A sample shuttle transfer is implemented via a modified PP3006 transfer solution from *Quorum Technologies Ltd, 2021*. The optical microscope hardware resides on a high vacuum (HV) door replacing the original door of the microscope, and it current design fits on a Thermo Fisher Scientific (TFS) SDB chamber. By adapting the design of the door, it will be compatible with dual beam systems retaining 52 degrees between SEM (top) and FIB (side) from other manufacturers. The piezo stages, microcooler, optical microscope, and interface to the host system are controlled using the open-source acquisition software Odemis (Delmic B.V.) (*Piel et al., 2022*). In the following sections, we will discuss the optical microscope, cryogenic sample holder, and positioning stage in more detail.

## Optical microscope

To allow optical inspection when the sample is in position for FIB milling at 10° angle of incidence, the OL is in a tilted position to retain perpendicular incidence to the sample surface (*Figure 2*). It resides in the HV chamber, along with the folding mirrors. All other optical components are mounted inside the optical module at ambient conditions and are separated by an optical vacuum window.

As the sample is held at cryogenic temperature, a relatively long working distance (WD) OL is required. We selected a Nikon CFI L Plan EPI 100xCRA objective lens (Nikon #MUE35900, 100×) for its intermittent NA of 0.85 and WD ranging between 0.85 to 1.2 mm depending on correction collar setting (0 to 0.7 mm). In our specific case (coverslip #1.5H, 0.17(5) nm thick) the WD is approximately 1 mm. Residual gas analysis with the OL in a separate vacuum setup did not show any foreign species due to the addition of the OL and we did not observe the imaging performance to deteriorate after repetitively pumping and flushing cycles.

In addition to four fluorescence channels, the setup also allows for imaging with reflected light microcopy (RLM) by collecting light reflected by the sample from either of the excitation channels through an additional pass-through hole in the filter wheel. Overlaying the FM and RLM images facilitates targeted milling by combining fluorescence signal with contextual information (*Figure 2e*).

Astigmatic point spread function (PSF) shaping is implemented by inserting two obliquely crossed cylindrical lenses, in the form of a Stokes variable power cross-cylinder lens set, in the detection path (*Stokes, 1849*; *Thompson, 1899*). One of the two lenses is mounted in a rotational mount, actuated by a stepper motor, and is used to switch between astigmatic and regular widefield imaging.

To validate the performance of the optical microscope, we recorded PSFs at cryogenic conditions by imaging and averaging fluorescent beads (sample temperature 300 K, $N = 20$, $\lambda = 520 \, \text{nm}$). The full width half maximum of the widefield PSF is 370 and 1000 nm for the $XY$ and $Z$ directions, respectively. The astigmatic PSF is recorded by setting $\theta = 92°$, which leads to an astigmatic deformation in the PSF, *Figure 2d*.

## Cryogenic sample holder

Conventional cryogenic FIB-SEM systems often use a cold-gas cooled sample holder consisting of a solid block of copper with positioning stage mechanics below it, which is incompatible with our design for the optical microscope. Our cryogenic sample holder (*Figure 3*) needs to: (1) keep the sample vitrified, (2) allow the sample to be imaged from three sides simultaneously by SEM, FIB, and LM, (3) accept TEM grids clipped in AutoGrid cartridges, (4) keep drift and vibration levels at a minimum, and (5) shield the sample from contamination as much as possible. We opted for a cold-finger design where the central feature is a customized Joule–Thomson (JT) cryogenic microcooler optimized for its low vibrations, drift and small footprint (DEMCON kryoz) (*DEMCON-kryoz, 2021*; *Lerou et al., 2008*), see *Figure 3c*. The cooling mechanism is based on high pressure gas undergoing JT expansion

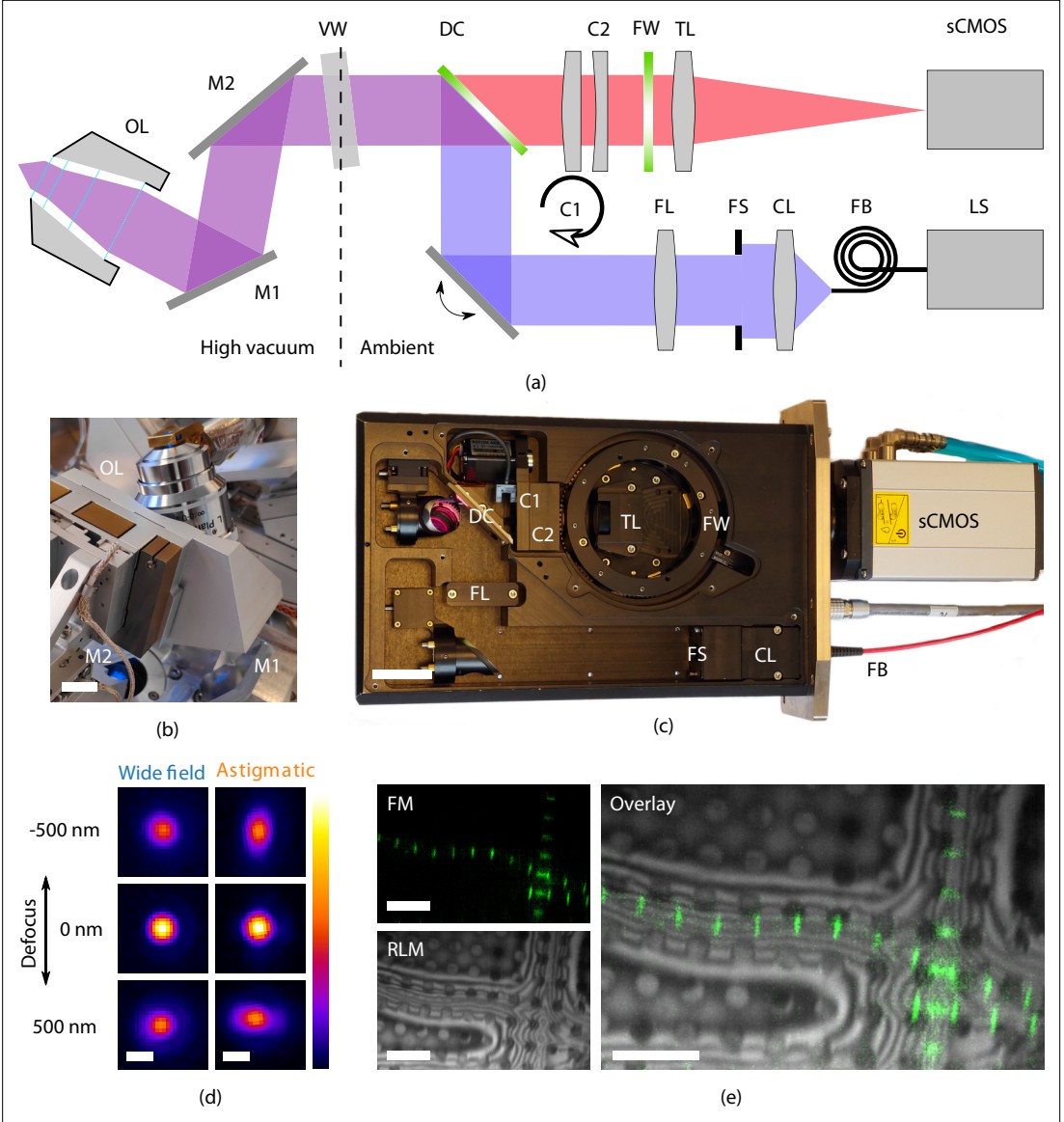

**Figure 2.** High-resolution fluorescence and reflected light microscopy allows to overlay fluorescence markers on sample structural layout. (**a**) Schematic illustration showing the layout of the epifluorescence microscope. $M1$ and $M2$ form a folding mirror residing in high vacuum together with the objective lens (OL). The vacuum window (VW) separates the high vacuum from ambient conditions. Excitation and emission paths are separated by the dichroic mirror (DC). Two cylindrical lenses ($C1$ and $C2$) are placed in the emission path to allow for astigmatic point spread function shaping. Emission filters are mounted in the filter wheel (FW), and the tube lens (TL) focuses the image on the camera (sCMOS). The light source (LS) is coupled by a fiber (FB) to the optical module. Light emitted from the fiber is collected by the collector lens (CL), clipped by the field stop (FS) and imaged into the OL object plane by the field lens (FS). (**b**) Optical OL mounted on 3DOF stage along with $M1$ and $M2$. Scale bar 1 cm. (**c**) Annotated photograph showing the optical module. Scale bar 4 cm. (**d**) $XY$ slices from point spread functions of the optical microscope, at different levels of defocus, both in case of widefield imaging and with induced astigmatic shaping. The point spread functions have been obtained by imaging and averaging individual fluorescent beads ($T_{sh} = 300\,\mathrm{K}$, $N = 20$, $\lambda = 520\,\mathrm{nm}$). Scale bar 0.5 µm. (**e**) Imaging modalities of the optical microscope. Left top: fluorescence microscope (FM) image of *Drosophila* myofibrils where Sls(700) is tagged with Alexa Fluor 488. Left bottom: reflected light microcopy (RLM) image of the same area as the FM image. Right: overlay of the FM and RLM images. The circular pattern originates from the Quantifoil R 2/2 film and interference fringes are visible along the radial direction of the myofibrils due to the changing thickness. Scale bar 8 µm.

through a restriction, producing a maximum net cooling power of ~200 mW. The cold stage consists of three layers of patterned borosilicate glass (D263T) housed in a titanium tube, and is braided to a cold finger. The 0.1 mm thick titanium tube provides adequate thermal isolation from the bigger aluminum body containing incoming gas lines, electrical wiring, and a vacuum connection. The housing is leak tight and differentially pumped by the FIB-SEM vacuum system through a fluoropolymer tube.

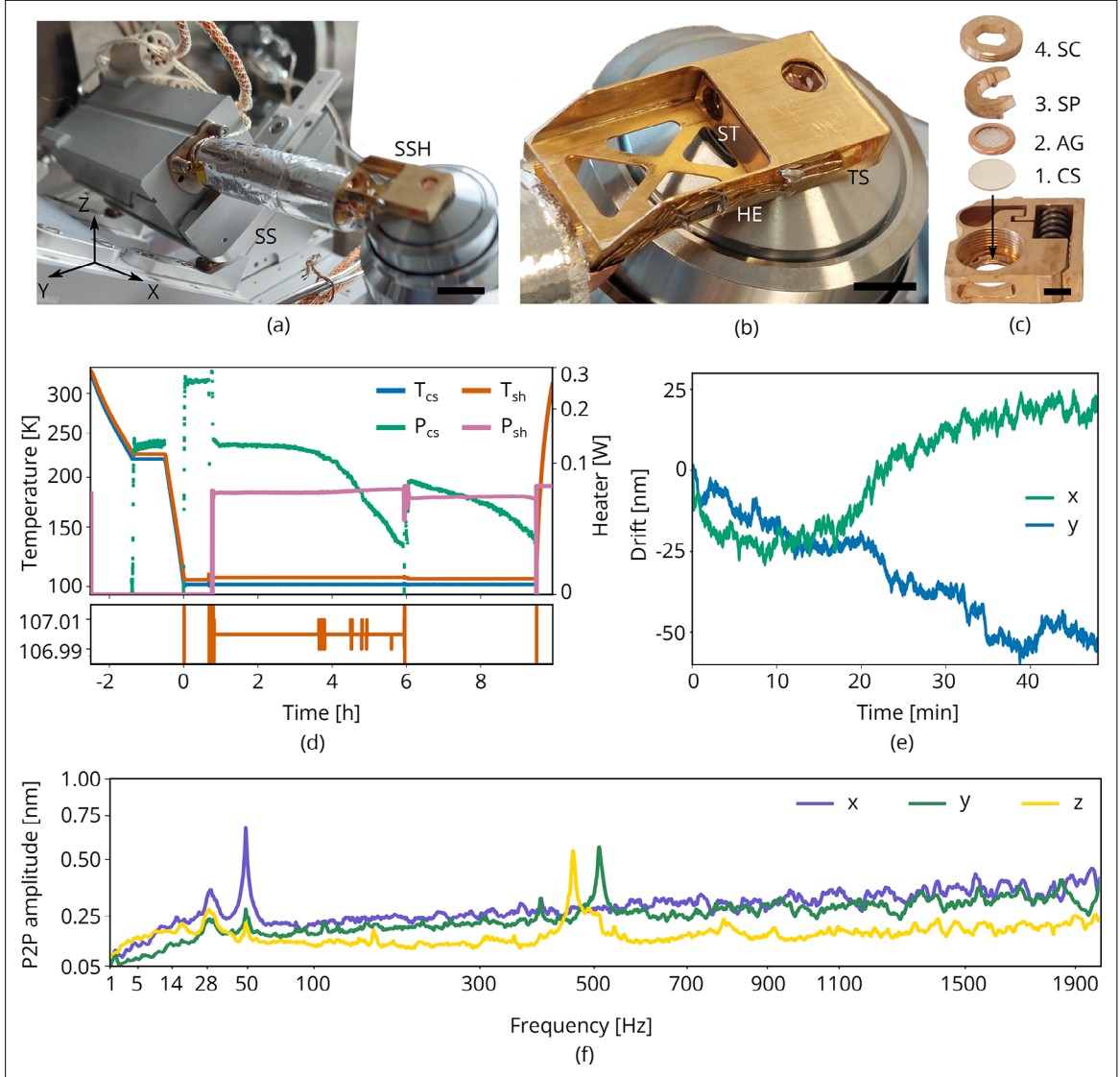

**Figure 3.** Cryogenic microcooler design and performance. Photographs of (**a**) the microcooler mounted on the sample positioning stage (SS), (**b**) the sample shuttle holder (SSH) in detail, and (**c**) the shuttle (ST) and the stacking order of the coverslip (CS), AutoGrid (AG), spacer ring (SP), and screw (SC). Scale bars 10 mm, 5 mm, and 2 mm, respectively. (**d**) Microcooler thermal performance over time. $T_{cs}$ and $T_{sh}$ refer to the cold stage and shuttle holder temperatures (101.5 K and 107 K, respectively, when stabilized). The heater power required for temperature stabilization is also plotted ($P_{cs}$ and $P_{sh}$). The shuttle holder temperature is stabilized within 10 to 20 mK as shown at the bottom. As cryopumping water from the rest gas slowly increases the radiative heat load, we changed the set point to 106 K at $t = 6$ h to have more heater power available for temperature stabilization. (**e**) Drift measured whilst regulating both cooler- and shuttle holder temperatures. (**f**) Peak-to-peak vibration amplitude as a function of frequency at 101 K cooler temperature ($T_{cs}$) and different directions. These spectra were measured with the microcooler mounted on an aluminium dummy stage inside the focused ion-beam scanning electron microscope (FIB-SEM) system. The peak at 50 Hz originates from electrical interference.

This compact cooling solution provides local cooling to the sample. We found that it requires approximately 2 hr to reach <108 K and the cooler can keep the sample cold for more than 9 h. The standing time is solely limited by cryopumping water from residual gas onto the cold surfaces as seen by the decrease in heater power ($P_{cs}$ and $P_{sh}$) required for temperature stabilization, see *Figure 3d*. The lack of any liquid boil-off keeps the vibration- and drift levels very low. With the shuttle holder temperature stabilized within 10 to 20 mK (*Figure 3d*, bottom), the drift levels are approximately 1 nm/min along the $y$ direction and 0.5 nm/min for the $x$ direction (*Figure 3e*). The peak-to-peak vibration levels are well below 1 nm in all directions (*Figure 3f*). Peaks around 500 Hz are caused by high pressure gas undergoing expansion through a restriction in the cold stage. An internal braid decouples these vibrations best along the $X$ direction, but is stiffer along $Y$ and $Z$ directions.

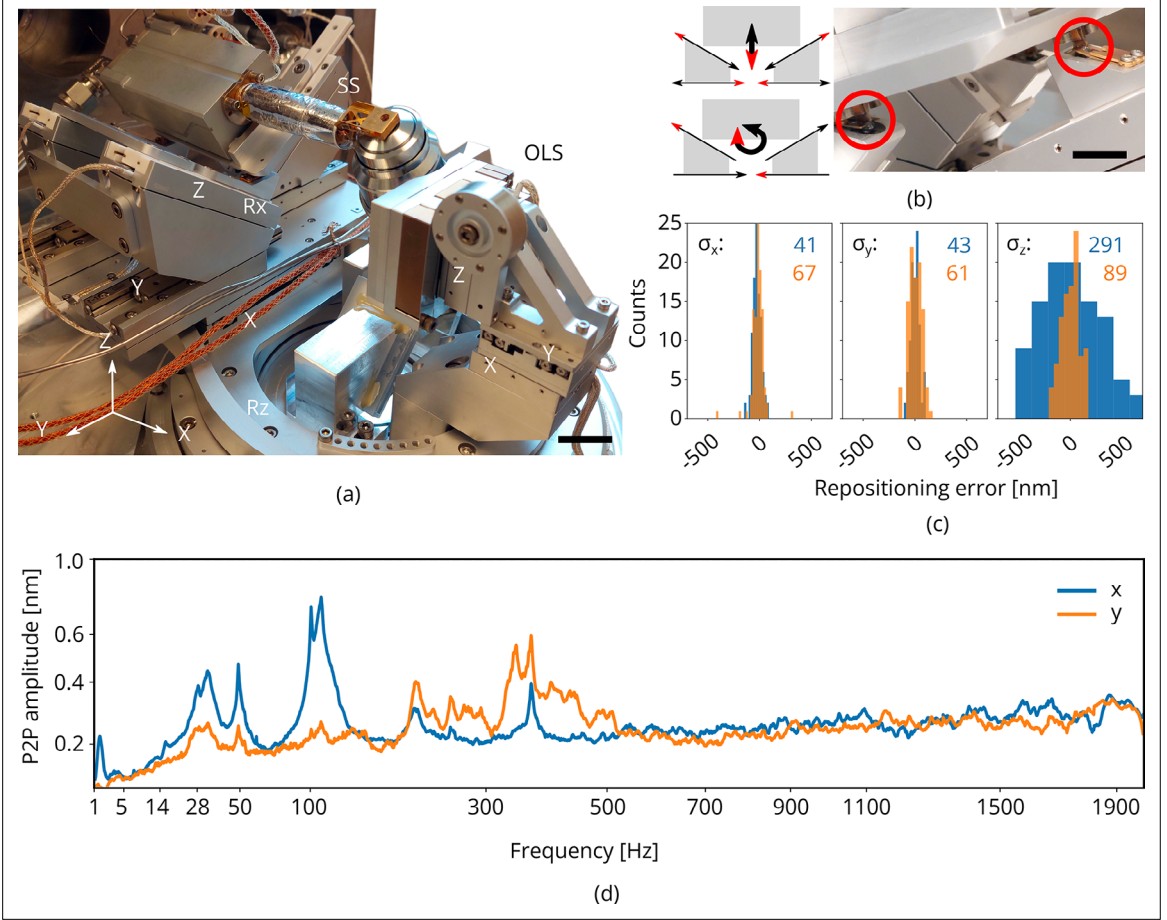

**Figure 4.** Piezo positioning stage design and performance. (**a**) Photograph of the piezo positioning stages used in the setup. The 5 degrees of freedom (DOF) sample stage (SS) moves the microcooler including sample holder around and the 3DOF stage (OLS) positions the OL. Scale bar 2 cm. (**b**) Schematic showing kinematic geometry for movements along $Z$ and $R_x$. The microcooler is mounted on an interface plate having three ceramic ball contacts and mount to three kinematic mounts; two v-grooves and one point contact. Scale bar 1 cm. (**c**) Repositioning error as measured when using the light microscope (LM) to repetitively centering and focusing the same feature in reflected light microcopy (RLM). Images of the sample are acquired with the scanning electron microscope (SEM) and focused ion beam (FIB) and the image-to-image shifts are used to determine the repositioning error in the $X$, $Y$, and $Z$ directions (sample coordinate system). $\sigma_{X,Y,Z}$ indicated in blue and orange in nanometer. Number of measurements $N = 100$. (**d**) Peak-to-peak vibration amplitude as a function of frequency with the microcooler mounted on the 5DOF stage inside the FIB-SEM system. The peak at 50 Hz originates from electrical interference. Due to reduced stiffness of the sample stage (as compared to the aluminum dummy stage), an overall increase is measured whilst staying well below 1 nm peak-to-peak.

## Piezo positioning stages

To position the sample within the vacuum chamber, the cryogenic microcooler is mounted on an in-vacuum piezo positioning stage having 5 degrees of freedom: $X$, $Y$, $Z$, $R_x$, and $R_z$. It is built up from individual linear crossed roller bearing positioners using a piezo stick-slip mechanism (SLC Series, SmarAct GmbH) (*Figure 4a*; *GmbH, 2021*). It is designed to have ±3 mm translation range in $X$, $Y$, $Z$ and ±28° rotation range in $R_x$ when aligned to the FIB-SEM coincident point. By setting $R_x = -28°$, the FIB maintains a 10° grazing incident angle to the sample surface. The stage has a clear aperture for the optical path and OL with the corresponding piezo positioning stage having 3 degrees of freedom: $X$, $Y$, $Z$.

The $X$ and $R_z$ axes of the sample stage operate independently whilst the $Y$, $Z$, and $R_x$ axes are coupled through a kinematic mount. The $Y$ positioners are coupled to three positioners mounted on wedges, rotating them 45° in $YZ$ plane. This geometry was optimized for its stiffness as rotations in $R_x$ require the microcooler and connected tubing to be moved about. $Z$ translation is obtained by moving the slanted positioners down whilst compensating the resulting $Y$ shift with the $Y$ positioners, see *Figure 4b*. In the detailed photograph on the right the microcooler mounting plate is shown. It has

three ceramic balls on the bottom which mount into three kinematic mounts: two v-grooves and one point contact. Leaf springs on top keep the ceramic balls in place. The stage electronics are programmed such that all sample stage moves are performed in the (possibly rotated) sample coordinate system.

We measured the repositioning error of the stage by initiating a random move of the sample positioning stage and then returning to the start position (*Figure 4c*). The same feature was brought into focus with FM (manually, by eye), and centered in the field of view (FOV). We then acquired images of the sample with the SEM and FIB. This procedure was repeated a number of times and the image-to-image shifts were used to determine the relative error in the $X$, $Y$, and $Z$ directions (sample coordinate system). From these measurements, we estimated $\sigma_X$, $\sigma_Y$, and $\sigma_Z$ to be 41, 43, and 291 nm, respectively, (blue) when focusing using a widefield PSF. When using an astigmatic PSF (~110 m$\lambda$, orange) the repositioning error along $Z$ is reduced significantly to 89 nm. To probe robustness to mechanical vibrations, we recorded the vibration spectrum at room temperature with the microcooler mounted

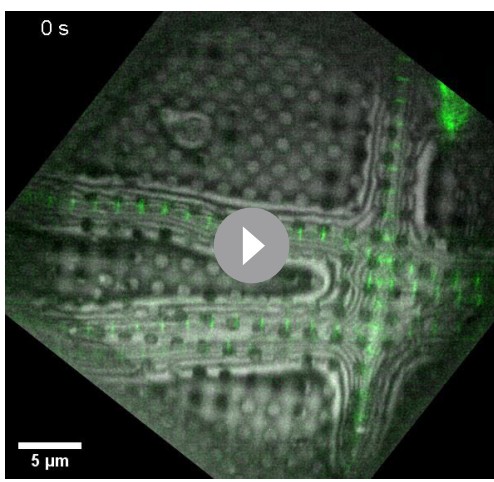

**Video 1.** Live LM imaging whilst milling. Movie showing RLM (grayscale) and FM (green) images recorded whilst FIB milling a frozen hydrated lamella. The 3-beam coincident geometry allows FM and RLM images to be acquired during lamella milling in a *Drosophila* myofibril sample. The fluorescence from Sls(700) tagged with Alexa Fluor 488 (green) is overlaid on the reflected light (grayscale).

https://elifesciences.org/articles/82891/figures#video1

on the sample stage. The reduced stiffness of the sample stage alters the vibration spectrum, showing more and broader peaks above 100 with respect to the aluminum dummy stage (*Figure 4d*). However, the peak-to-peak vibration levels are well below 1 nm.

## Light microscopy-assisted lamella fabrication in vitrified cells
### Sample loading and three-beam alignment
With the microcooler at its operating temperature (<108 K), it is moved into the loading position by the piezo stage. Plunge frozen vitrified cells on a clipped TEM grid are loaded in the sample shuttle. The sample shuttle is picked up and transferred in using the modified Quorum transfer system. The sample stage is then positioned such that the TEM grid is brought into the SEM and FIB focus by positioning it near the coincident point of the FIB-SEM. Next, the OL stage is used to bring the sample into the optical microscope focus and care is taken to align the three beams by imaging a piece of bare grid foil with all imaging modalities. The thin (15 to 50 nm) grid foil allows precise alignment of all three beams in the following way: first the sample $Z$ height is adjusted such that SEM and FIB image the same area on the sample. Next, the OL position is adjusted such that this area is also imaged in RLM and hence coincidence is achieved with the three beams. We found that the stability of the coincidence alignment was approximately 0.7, −0.2, and −0.05 µm/h for the $X$, $Y$, and $Z$ directions, respectively. The largest drift is observed along the lamella width ($X$) which is typically not the most critical dimension, as its width usually spans several micrometers. See the Materials and methods for more details.

After this, the OL stage is not moved anymore and sample navigation is done solely with the sample positioning stage ($XY$). Bringing an ROI into the optical microscope focus is also done with the sample stage ($Z$) as $X$, $Y$, and $Z$ moves are performed in the rotated coordinate system of the sample. Having an ROI in focus and centered in the FOV of the FM/RLM automatically aligns it to the center of the FIB and hence milling can commence.

### Live monitoring and adjustment of FIB milling
FM and RLM images acquired before and during the milling process were used to monitor FIB milling (*Video 1*). *Drosophila* and zebrafish myofibrils samples were used for these experiments (Sls(700) tagged

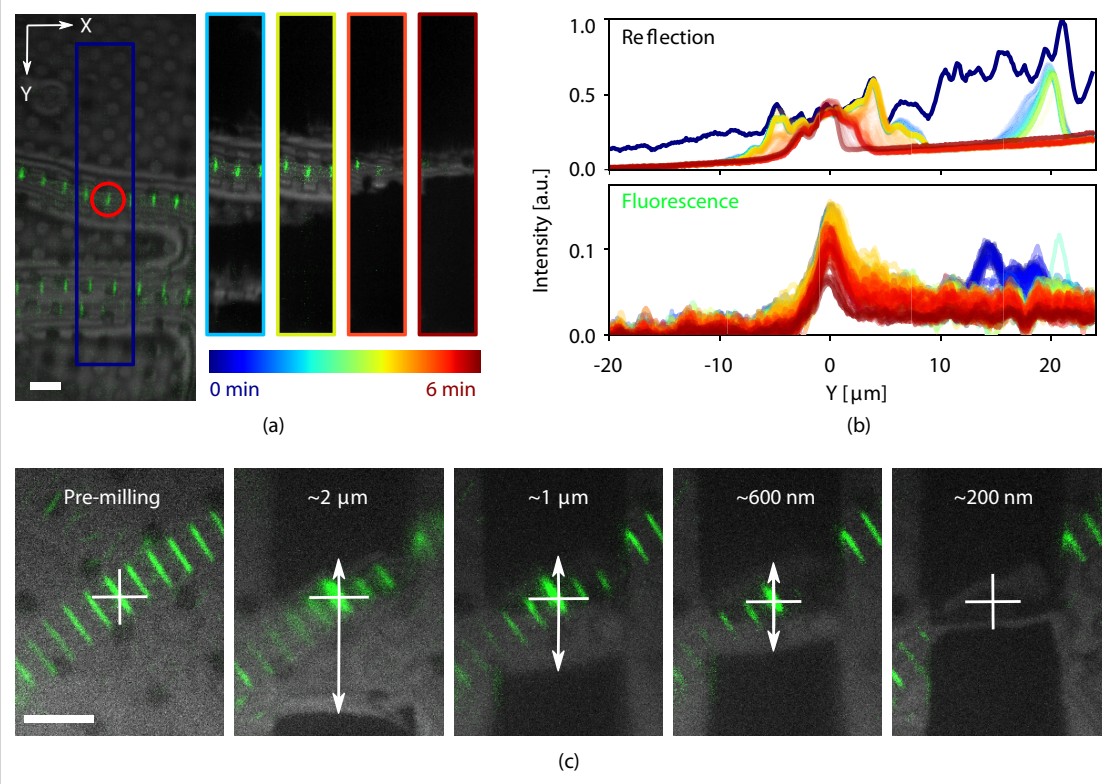

**Figure 5.** Light assisted focused ion beam (FIB) milling of frozen-hydrated lamellae. (**a**) Reflected light microcopy (RLM) and fluorescence microscope (FM) images were recorded whilst milling. The image on the left shows the indicated milling area marked in dark blue. The four images to the right show snapshots at different stages of milling, denoted with the varying color scale. This color scale is also used in (**b**) where the normalized integrated pixel intensity across the milling area $Y$ direction is shown for both the reflection and fluorescence channels. (**c**) Stepwise milling progression as observed by RLM and FM. Estimated lamella thickness based on FIB images indicated at the top of the images. The fluorescent feature of interest is marked by the white cross and any misalignment (white arrows) can be corrected for. All scale bars are 5 µm. *Drosophila* and zebrafish myofibrils in (**a**) and (**c**), respectively. Fluorescence marks the Z-disc of the sarcomere.

with Alexa Fluor 488). The fluorescence (green) is overlaid on the reflected light (grayscale). A few cropped snapshots from this live recording are shown in *Figure 5a*. The image on the left shows the milling area by the dark blue rectangle and the fluorescent target is red encircled. Four different snapshots of this milling area are shown on the right. The varying color scale is used to indicate the progress in time ranging from 0 to 6 min. In *Figure 5b*, the intensity of both the FM and RLM signals within the milling area is integrated along $X$ and normalized. The dark blue curves are the integrated pixel intensity prior to milling. The light blue peak on the right in the reflection plot corresponds to the second (bottom) myofibril visible in *Figure 5a*. As the lamella gets thinner, the central reflected intensity peak becomes narrower (going from yellow to red). As the fluorescent target is larger than the lamella thickness, some loss of fluorescence is expected as is shown in the bottom plot. Although the fluorescence signal decreases it still remains after milling. The full video can be found in the Supplementary information.

Quality control can be performed during every step of the milling procedure without the need for stage movements when switching between the different imaging modalities. Consequently, any misalignment of a feature of interest with respect to the center of the lamella is spotted and corrected for. After each step, RLM and FM images are acquired. With the feature of interest marked by the white cross, it is easy to spot a misalignment (white arrows) and correct for this by shifting the milling patterns accordingly. This prevents accidentally milling away the feature of interest and increases sample preparation yield.

## Fluorescence targeted milling and cryo-ET

We illustrate a fluorescence targeted lamella fabrication workflow by targeting autophagic compartments in HeLa cells (*Figure 6*). We used a red fluorescent protein (RFP)-GFP tandem fluorescent-tagged

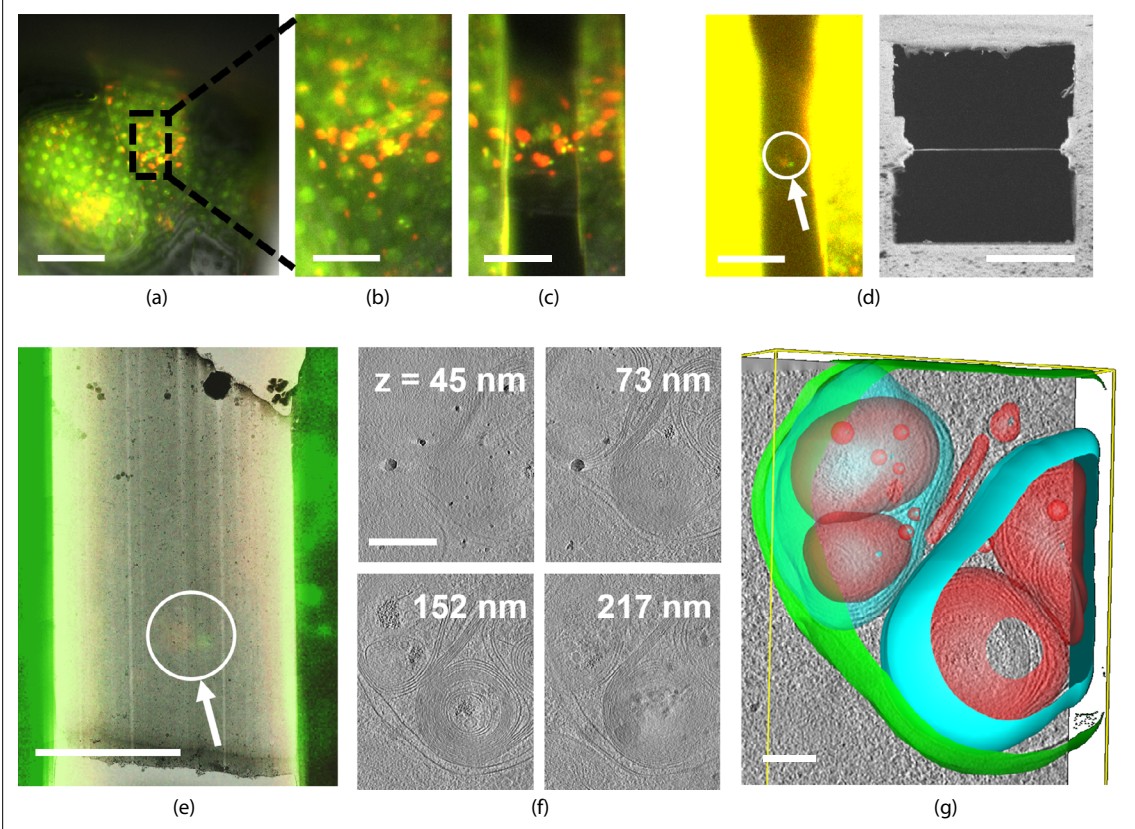

**Figure 6.** Light targeted focused ion beam (FIB) milling of frozen-hydrated lamellae. (**a**) First the sample is brought into the center of the field of view (FOV) and focused properly. Green and yellow: autophagosomes and red: autophagolysosomes. (**b**) Close-up of (**a**) showing the milling area in more detail. (**c**) Snapshot taken at an intermediate milling step. (**d**) Left: the lamella finished and polished, imaged in the fluorescence microscope (FM) and the contrast enhanced to show the feature of interest. Right: a snapshot acquired with the FIB showing the polished lamella. (**e**) The reflected light microcopy (RLM) channel is used to align the light microscope data with the transmission electron microscope (TEM) overview image. The fluorescent feature of interest indicates where the tomogram should be acquired. (**f**) $Z$ slices of the reconstructed TEM tomogram with (**g**) segmented membranes. Annotated are autophagolysosome outer membrane (green), autophagolysosome membraneous content (blue and red). Scale bars: (**a**) −40 µm, (**b**) −10 µm, (**c**) −5 µm, (**d**) left −10 µm and right −5 µm, (**e**) −5 µm, (**f**) −0.25 µm, and (**g**) −0.1 µm.

LC3 (mRFP-GFP-LC3) single molecule-based probe that can monitor the autophagosome maturation process (*Kimura et al., 2007*). This probe emits yellow signals (GFP plus RFP) in the cytosol and on autophagosomes but only red signals in autolysosomes because GFP is more easily quenched and/or degraded in the lysosome than RFP (*Katayama et al., 2008*). We selected regions of LC3-positive punctae using the mRFP-GFP fluorescence signal and focused this region in the center of the FOV. We then cut the lamella step-by-step and finally polished whilst making sure the feature of interest is still present (*Figure 6a–d*) at all stages. The contrast was increased to make the fluorescence visible (*Figure 6d*, left) and the final lamella imaged by FIB in the right panel. The sample containing the milled lamella was removed from the FIB-SEM using the transfer system and loaded into the auto-loader of the cryo-TEM. An overview image of the lamella was acquired and using the RLM image data we overlaid the FM signal guided by the outline of the lamella as imaged by TEM and RLM. We then acquired tilt series at the sites still showing fluorescence and a z-stack is reconstructed the tomo-grams. No signs of devitrification were visible in these reconstructions, indicating the sample has been well preserved during the entire procedure.

## Discussion

Currently, the system presented is solely compatible with samples in an AutoGrid form factor. With minor modifications, other planar sample types, such as samples prepared by the Waffle method (*Kelley et al., 2022*), would fit inside the sample shuttle. Identifying grid bars and sample navigation

can be done directly using the LM. Imaging through a relatively thick sample (~50 μm) pose potential challenges, as the increased imaging depth is likely to induce aberrations and increased scattering.

Although the system presented reduces the sample handling steps, contamination and avoids the use of fiducial markers, it also has its limitations: (1) Without an additional external cryo-FM, FM can only be done when the sample is clipped and loaded into the FIB-SEM; (2) A required WD of ~1 mm prevents high (>1) NA OLs to be used; (3) Implementing a cryo-immersion OL adds significant complications and is not straight forward; (4) The integrated approach reduces the number of available methods for super-resolution microscopy due to the need for a compact optical path.

External cryogenic super-resolution microscopy can readily provide subdiffraction limit imaging and can provide a milling target, but the localization precision will only be as good as the correlation accuracy provided by the fiducial markers images in both LM and FIB. This same limitation also applies to the different commercially available integrated LM systems that can be retrofitted onto the FIB-SEM, but require in-vacuum stage movements (*ThermoFisher Scientific, 2021*; *Smeets et al., 2021*).

We have implemented a system of two rotatable cylindrical lenses to allow fluorescence imaging with an astigmatic PSF. We have demonstrated that astigmatic imaging improves relocalization accuracy after stage movements to within 90 nm along the optical axis. Further work could push this concept further toward quantitative localization of multiple or potentially even single fluorophores and consequently targeted milling with even lower positional errors. This would also open up the possibility for automated milling, based on acquired LM (astigmatic) data. For this to work, we would need to automatically correct for deviations in the coincident alignment or update the alignment regularly, but this requires more work to be done.

Moreover, an intriguing prospect would be to extend this concept toward cryogenic super-resolution microscopy correlated with both SEM and FIB. Potentially, this could also allow for super-resolution FM imaging after polishing the lamella, ultimately combining at the single protein level high-resolution biological information with the structural data obtained from TEM.

## Conclusion

We have developed a coincident three-beam microscope for the cryo-ET workflow which allows direct light microscopy targeted lamella fabrication, without the need for repositioning or fiducial markers. The fluorescence signal intensity from the lamella can be monitored whilst milling making sure that the target remains intact. Any misalignment can be directly observed and corrected. The five-channel inverted epifluorescent LM is diffraction limited in widefield imaging with an NA up to 0.85. Astigmatic PSF shaping is achieved through the use of a Stokes variable power cross-cylinder lens set. This allows positioning of the sample for targeted milling with errors as small as 25 nm in $XY$ and 90 nm in $Z$. The system utilizes a novel, liquid nitrogen-free cooler design having low levels of vibrations, drift, and an up time of more than 9 hr. The entire cryogenic FM can be integrated into a regular FIB-SEM, effectively converting it into a cryo-LM-FIB-SEM, making it possible to add in situ fluorescence targeted milling in an established cryo-ET workflow.

Our approach allows for a live, reflection and/or fluorescence image feed while milling takes place. This opens the additional possibility of an automated milling procedure in which the fluorescent feature of interest is with certainty inside the lamella, before the sample is transferred to the TEM.

## Materials and methods
### Epifluorescence microscope

Excitation and emission paths are separated by a dichroic mirror (Semrock #FF410/504/582/669-Di01-25x36), matched by four separate emission filters (Semrock #FF01-440/40-25, #FF01-525/30-25, #FF01-607/36-25, #FF02-684/24-25) in a filter wheel. A 200 mm tube lens (TL) is placed in the center of the filter wheel, and the camera (water cooled Andor Zyla 4.2 PLUS) is fitted on the outside of the optical module. The optical window (Edmund #16-461) separating the HV and ambient environments is mounted at 87° to the optical axis, avoiding a parasitic reflection from excitation light on the camera sensor. A fiber-coupled light emitting diode (LED) excitation light source (Lumencor SPECTRA X central excitation wavelengths: 390, 485, 560, and 648 nm) is used. Light emitted by the fiber is collected by an aspheric collection lens (Edmund #66-312). Together with the field lens (Qioptiq #G322342322), the fiber output is imaged into the exit pupil of the OL. A field stop is placed between

the collection and field lenses and is imaged onto the OL object plane, thus restricting the illuminated area on the object (sample). Alignment of the optical path is done by adjusting the 3 DOF kinematic mirror mounts in the excitation and emission paths.

The Stokes variable power cross-cylinder lens set is composed of a plano concave and a plano convex round cylindrical lens (Thorlabs #LK1002RM-A, $f = -1000$ mm and Thorlabs #LJ1516RM-A $f = 1000$ mm). If the power axes of the two lenses are positioned at right angles ($\theta = 90°$), no astigmatism is introduced and hence the PSF remains unmodified. Consequently, variable amounts of astigmatism can be introduced by setting $\theta \neq 90°$.

## Microcooler shuttle holder

The sample shuttle holder mounts to the cold finger protruding at the front of the microcooler. Two temperature sensors (Pt-1000) are present, one residing on the shuttle holder ($T_{sh}$) and one placed on the micro cold stage ($T_{cs}$). Near each sensor, a surface-mount technology-based resistor (68 $\Omega$, International Manufacturing Services #IMS004-3-68) is placed acting as a heater. The thermal gradient between shuttle holder and micro cold stage is ~5 K, allowing the temperature of both to be regulated semi-independently through control loops. The shuttle contains the AutoGrid (Thermo Fisher Scientific) in which the TEM grid is held in place by a c-clip. Starting with an empty shuttle, first the coverslip (ITO-coated D263M glass, 3.4 mm diameter, Schott AG) is placed, followed by the Auto-Grid, spacer ring and screw. The shuttle has an actuated leaf spring mechanism which is counteracted when the shuttle is picked up by the transfer rod via a threaded interface. With the transfer rod fully engaged, the leaf spring folds around the main body to reduce the shuttle width so it can move inside the shuttle holder. When unscrewing the transfer rod, the compression spring pushes the leaf spring mechanism outwards, effectively clamping the shuttle inside the holder to ensure good thermal contact between holder and shuttle.

## PSF analysis

Multicolor fluorescent beads (150 ± 30 nm, #FP-0257-2, Spherotech Inc) were drop casted at low concentration on the glass coverslip. With a multitude of beads ($N > 20$) visible in the FOV, a z-stack was acquired (−3.5 to 3.5 μm, step size of 50 nm, $\lambda = 520\,nm$, sample temperature 300 K). From this stack, the PSF was measured by averaging multiple bead images using a Python code, as available on GitHub (*Boltje et al., 2022*). This code works in the following way. The z-stack is loaded as an array and a maximum intensity project along $z$ is calculated. Next, TrackPy is used to find all potential features in the 2D image (*Allan et al., 2021*). The features found are filtered on minimum- and maximum mass to discard noise peaks and clusters of multiple beads such that only single fluorescent beads remain. Beads too close to the edge or to each other are filtered out and the remaining beads are extracted in their respective subvolumes, based on the NA, the emission wavelength and the pixel size. Any remaining outliers (i.e., doublets or faulty localization) are filtered out by computing the Pearson correlation coefficient between the maximum intensity projections along $z$ of all subvolumes. To localize the remaining beads with subpixel precision a 2D Gaussian fit (nonlinear least squares) is performed on the subvolume maximum intensity projection along $z$ to get the $x$ and $y$ positions. At this coordinate, a 1D Gaussian fit is performed on the line intensity profile along $z$. Finally, the subvolumes are upsampled by a factor of five, the beads are aligned based on the individual $x$, $y$ and $z$ coordinates rounded to the closest upsampled voxel and are averaged yielding a close approximation of the PSF of the optical system. To obtain an astigmatic PSF, astigmatism was introduced by setting the cylindrical lens rotation to $\theta = 93.7°$, and the procedure for PSF extraction was repeated.

## Repositioning accuracy

The sample stage is positioned such that both SEM and FIB image the metal grid bar on an EM grid. The LM is aligned to image fluorescent beads (150 ± 30 nm, #FP-0257-2, Spherotech Inc) on the glass coverslip and a single isolated bead is centered in the FOV of the LM. The position is read from the stage encoders and stored. The sample stage is then ordered to make a random 3D move with limits of ±100 and ±30 μm for $XY$ and $Z$, respectively, and moved back to the previous position. The LM is used to manually center and focus the same fluorescent bead in the FOV and after this, images in the FIB and SEM are acquired. This procedure is repeated for a total on 100 times.

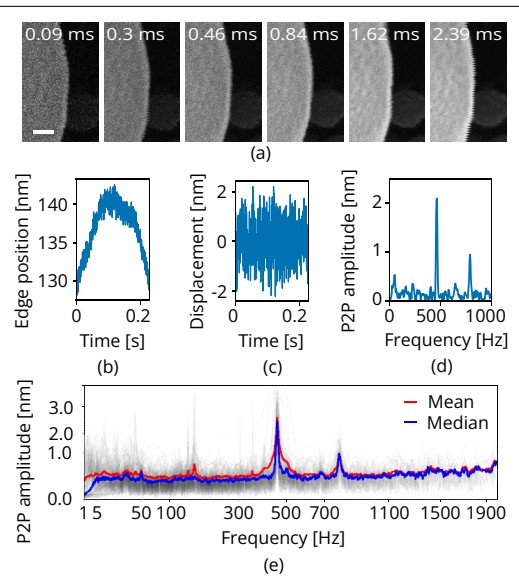

**Figure 7.** Scanning electron microscope (SEM) vibration analysis procedure. (**a**) A series of SEM images acquired with different dwell times and thus line times (the latter indicated at the top). (**b**) The edge position as a function of time for one of these images. (**c**) The vibration displacement as a function of time. Finally in (**d**), the scalloping loss corrected fast fourier transform (FFT). Processing all images acquired with the same scan rotation finally yields (**e**). Scale bar 25 nm.

The image-to-image shifts are determined by phase cross correlation (as implemented in the Skimage.registration module), for the sequence of FIB and SEM images separately. Due to the viewing geometry (62 and 10° angle of incidence to the sample plane for SEM and FIB, respectively) shifts along the image $Y_i$ direction can be due both sample displacement along $Y_s$ and $Z_s$. We extract the sample displacement by minimizing the square error between the two image shifts found ($Y_{SEM}$ and $Y_{FIB}$) and computed image shifts based on the imaging geometry. This yields 99 displacements along the $X$, $Y$, and $Z$ sample coordinate system representing the sample stage repositioning accuracy.

## Vibration measurements

Mechanical vibrations appear when scanning along an edge with clear contrast in the SEM, as is often done to gauge vibration performance (*Jung et al., 2012*; *Płuska et al., 2009*). We use a tin on carbon resolution test sample (#AGS1967T, Agar Scientific Ltd). As the line time determines the sampling frequency we acquire a series of images with different scan resolutions and pixel dwell times, effectively sampling a frequency range of approximately 0.1 to 5000 Hz (*Figure 7a*). This is done at 0 and 90° scan rotation to measure along the $X$ and $Y$ directions. Each of the images is analyzed line-by-line where a Heaviside step function is used to determine edge position. The Pearson correlation matrix is computed between the original image and a matrix of Heaviside functions having all possible step positions along the horizontal image direction. The vectorized Pearson correlation implementation is used from the GitHub repository belonging to *Enache et al., 2018*. The step position versus time can be extracted from the Pearson correlation matrix by taking the absolute value and looking up the indices of the maximum values (*Figure 7b*). The tin ball curvature is removed by applying a Butter high pass filter having a critical frequency $f_c = 5/T_f$ where $T_f$ is frame time of the image (*Figure 7c*). A scalloping loss corrected FFT is computed, having correct peak-to-peak amplitudes (*Lyons, 2011*; *Figure 7d*). 100+ images are acquired (Helios Nanolab 650 dualbeam, 18 kV, 100 pA beam current, 100 nm horizontal field width, UHR mode) and analyzed for each scanning direction, the median of the resulting FFTs is taken to come to the final vibration spectrum (*Figure 7e*). The Python code is available on GitHub (*Boltje and Last, 2022*).

## Three-beam alignment and fluorescence targeted milling in cryo-FIB-SEM

Validation experiments using the system have been carried out at the Department of Structural Biochemistry, Max Planck Institute of Molecular Physiology in Dortmund. The original HV door of the Aquilos Cryo-FIB (Thermo Fisher Scientific) was replaced with our HV door, after which the system was evacuated and pumped for several days. Prior to experiments, the preparation station in the low-humidity glove box, the cryogenic chamber shield and the microcooler were cooled down (*Tacke et al., 2021*). Cellular samples on clipped EM grids were mounted into the shuttle inside the glove box, picked up with the modified Quorum transfer system and transferred into the system via a load lock. After loading, the sample stage was homed and moved to the coating position, where the gas injection system is used to apply a Pt precursor coating to the EM grid through the top SEM access

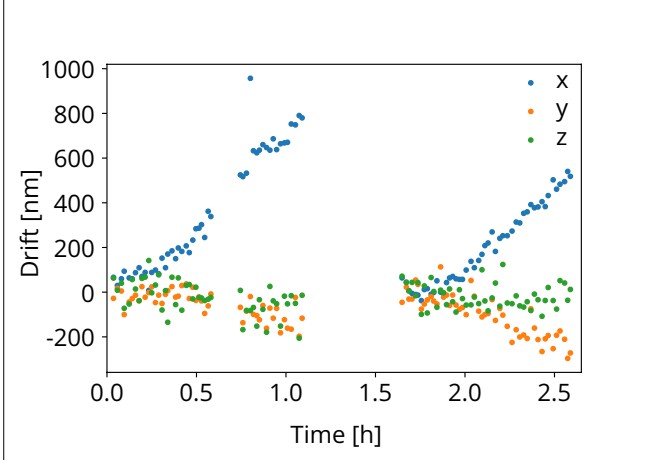

**Figure 8.** Objective lens (OL) stage drift as a function of time as measured during the stage repositioning moves.

hole in the shuttle. Afterwards, the sample stage is moved to the predefined three-beam imaging position, followed by the OL stage.

Due to the large distance traversed, the three-beam alignment needs to be refined after loading a sample, and is done by imaging a piece of bare grid foil with all imaging modalities. The thin (15 to 50 nm) grid foil allows precise alignment of all three beams: (1) The sample $Z$ height is adjusted such that SEM and FIB image the same area on the sample (i.e., eucentric height, the sample is in the coincident point of the SEM and FIB beams). (2) The OL position is adjusted such that this area is also imaged in RLM and hence coincidence is achieved with the three beams. (3) A small hole (2 diameter) is milled through the grid foil, using the FIB. (4) This hole is used as an alignment feature, effectively refining the three-beam coincidence as outline above. This process takes about 5 to 10 min, and after this sample navigation and focusing is done through the sample stage.

To test the stability of the coincidence alignment, we computed the cumulative image shift for all FIB-SEM images acquired for the stage repositioning measurements. We obtain the sample shift along the $X$, $Y$, and $Z$ directions as explained above. While the sample positioning stage performs random moves, it is aligned to the focal point of the optical objective when FIB-SEM images are acquired. The objective stage position is *not* actively moved or controlled. Hence, any objective stage drift, would result in $XY$ shifts in the FIB-SEM images acquired. This is shown in *Figure 8* as a function of time. The gaps in the recording are due to a lunch and a coffee break. At 1.7 h, a new location on the sample was chosen to avoid bleaching of the fluorescent target bead. Along the $X$ direction the drift was approximately ~0.7 μm/h and for the $Y$ and $Z$ directions this amounts to −0.2 and −0.05 μm/h respectively.

Having an ROI in focus and centered in the FOV of the FM/RLM automatically aligns it to the center of the FIB. Sites for milling were identified by first identifying them in a low-magnification SEM overview image, followed by a more thorough inspection with the LM (both FM and RLM). With a selected ROI in focus and centered into the LM FOV, FIB milling can commence, starting with the first rough milling step along with the relief cuts.

The SEM is used to assess the lamella thickness, and when thin enough, a final inspection is done with the LM. With 3 to 6 milling sites complete, the sample was transferred out of the system and into the glovebox. The AutoGrid was taken out of the shuttle and placed into the autoloader cassette which was loaded into the NanoCab (Thermo Fisher Scientific) and transferred to the electron cryo-microscope.

## Sample preparation

*Drosophila* flight muscle myofibrils and zebrafish myofibrils were isolated as described previously by *Weitkunat and Schnorrer, 2014* and *Wang et al., 2021*, respectively.

HeLa cells stably expressing the mRFP-GFP-LC3 reporter construct (HeLa mRFP-GFP-LC3; kind gift from Fulvio Reggiori, University of Groningen, Netherlands) were cultured in Dulbecco's modified Eagle medium (DMEM), 10% fetal calf serum, 1% pen/strep supplemented with 0.6 μg/mL G418 at

37 °C with 5% $CO_2$ and a humidified atmosphere. Prior to seeding on grids, cells were washed three times with warm medium, followed by two washing steps with phosphate-buffered saline (PBS). 1 mL of trypsin was added, followed by a 5 min incubation at 37 °C, in a 5% $CO_2$ humidified atmosphere. Cells were seeded at 660,000 per tissue culture dish (60 mm; Sarstedt). After a brief cleaning with ethanol and prewashing with DMEM medium, microscopy slides were placed within the culture dish. Glow-discharged carbon-coated gold mesh grids (QF 2/2 AU 200, Quantifoil) were placed on the edge of the glass microscopy slides, and cells were gently dispensed into the dish. Cells were cultured for an additional 18 h and then washed twice with prewarmed PBS to remove any remaining culture medium. Then 5 mL of warm starvation buffer (20 mM HEPES, 4-(2-hydroxyethyl)-1-piperazineethanes ulfonic acid, pH 7.5, 140 mM NaCl, 1 mM $CaCl_2$, 1 mM $MgCl_2$, 10 mM glucose) was added. After 2 h at 37 °C, 5% $CO_2$, the cells were imaged using the EVOS cell imaging system (Thermo Fisher Scientific). Gold mesh grids were removed from the tissue culture dish and immediately prior to blotting, 3 µL of warm starvation buffer was added to the grid. Blotting conditions on the Leica GP2 vitrification robot were blotting from the back for ~2 s in a chamber with 98% humidity at 21 °C and then plunged into liquid ethane.

## Cryo-electron tomography and tomogram reconstruction

To avoid contamination, grids containing milled lamellae were transferred through a low-humidity glovebox (*Tacke et al., 2021*), into a Titan Krios transmission electron microscope (Thermo Fisher Scientific), equipped with a K3 camera (Gatan) and an energy filter. Projection images were acquired using SerialEM software (*Mastronarde, 2005*). Overview images of were acquired at 4800× or 8700× nominal magnification to identify locations for high-magnification tilt series acquisition. Tilt series were acquired at 42,000×, 53,000×, or 81,000× nominal magnification (pixel sizes 2.32Å, 1.81Å, and 1.18Å, respectively). A dose-symmetric tilting scheme (*Hagen et al., 2017*) was used during acquisition with a tilt range of −56° to 56° relative to the lamella plane at 2° increments. A total dose of 130 to 150 e–/Å$^2$ was applied to the sample.

Individual tilt movies acquired from the microscope were motion corrected, contrast transfer function corrected and combined into stacks using Warp (*Tegunov and Cramer, 2019*). The combined stacks were then aligned and reconstructed using IMOD (*Kremer et al., 1996*).

## Acknowledgements

*Drosophila* flight muscle myofibrils were kindly provided by EH Chan and F Schnorrer (Institut de Biologie du Développement de Marseille), and zebrafish myofibrils by Y Hinits and M Gautel (King's College London). We express our gratitude to Fulvio Reggiori (University of Groningen, Netherlands) for providing the HeLa cells and are thankful to Mingjun Xu for help during sample preparation. We thank Andries Effting (Delmic BV) for helpful discussions, and we would like to thank Ryan Lane (TU Delft) for his contribution in various Python developments. The majority of the 3D CAD design was done by Thomas van der Heijden (Delmic BV), for which we are grateful. This work was financially supported by NWO-TTW project no. 17152 to JPH, NIH grant RO1 AI127401 to GJJ, European SME2 grant no. 879673 to Delmic BV, Eurostars grant no. E13008 to SH & SR, and ERC grant ERC-StG-852880 to AJJ.

## Additional information

### Competing interests

Daan B Boltje: is an employee of Delmic B.V. Jacob P Hoogenboom, Sander den Hoedt: has a financial interest in Delmic B.V. Caspar TH Jonker, Mart GF Last: were employees of Delmic B.V. The other authors declare that no competing interests exist.

## Funding

| Funder | Grant reference number | Author |
|---|---|---|
| Nederlandse Organisatie voor Wetenschappelijk Onderzoek | TTW No 17152 | Jacob P Hoogenboom |
| National Institutes of Health | RO1 AI127401 | Grant J Jensen |
| European Commission | SME2 No 879673 | Sander den Hoedt |
| Eurostars | No E13008 | Stefan Raunser Sander den Hoedt |
| European Research Council | StG-852880 | Arjen J Jakobi |

The funders had no role in study design, data collection, and interpretation, or the decision to submit the work for publication.

## Author contributions

Daan B Boltje, Conceptualization, Data curation, Software, Formal analysis, Validation, Investigation, Visualization, Methodology, Writing – original draft, Project administration, Writing – review and editing; Jacob P Hoogenboom, Arjen J Jakobi, Conceptualization, Resources, Supervision, Funding acquisition, Project administration, Writing – review and editing; Grant J Jensen, Abraham J Koster, Jürgen M Plitzko, Stefan Raunser, Roger Wepf, Conceptualization, Supervision, Funding acquisition, Project administration, Writing – review and editing; Caspar TH Jonker, Validation, Investigation, Visualization, Methodology, Project administration, Writing – review and editing; Max J Kaag, Software, Investigation, Methodology; Mart GF Last, Data curation, Software, Formal analysis, Validation, Investigation, Visualization, Methodology, Writing – review and editing; Cecilia de Agrela Pinto, Resources, Methodology, Writing – review and editing; Sebastian Tacke, Formal analysis, Validation, Investigation, Methodology, Writing – review and editing; Zhexin Wang, Resources, Investigation, Methodology, Writing – review and editing; Ernest B van der Wee, Software, Formal analysis, Validation, Investigation, Writing – review and editing; Sander den Hoedt, Conceptualization, Supervision, Funding acquisition, Project administration, Writing – review and editing, Conceived and initialized the collaboration between the different research groups

## Author ORCIDs

Daan B Boltje  http://orcid.org/0000-0003-4881-4700
Arjen J Jakobi  http://orcid.org/0000-0002-7761-2027
Grant J Jensen  http://orcid.org/0000-0003-1556-4864
Abraham J Koster  http://orcid.org/0000-0003-1717-2549
Mart GF Last  http://orcid.org/0000-0002-3739-8863
Jürgen M Plitzko  http://orcid.org/0000-0002-6402-8315
Stefan Raunser  http://orcid.org/0000-0001-9373-3016
Zhexin Wang  http://orcid.org/0000-0002-4256-1143
Ernest B van der Wee  http://orcid.org/0000-0002-0139-4019

## Decision letter and Author response

Decision letter https://doi.org/10.7554/eLife.82891.sa1
Author response https://doi.org/10.7554/eLife.82891.sa2

## Additional files

### Supplementary files
• MDAR checklist

### Data availability
The data underlying the publication can be found at international data repository service 4TU.ResearchData, https://doi.org/10.4121/20787274.

The following dataset was generated:

| Author(s) | Year | Dataset title | Dataset URL | Database and Identifier |
|---|---|---|---|---|
| Boltje DB, Hoogenboom JP, Jakobi AJ, Jensen GJ, Jonker CTH, Kaag MJ, Koster AJ, Last MGF, de Agrela Pinto C, Plitzko JM, Raunser S, Tacke S, Wang Z, van der Wee EB, Wepf R, den Hoedt S | 2022 | Data underlying the publication: A cryogenic, coincident fluorescence, electron and ion beam microscope | https://doi.org/10.4121/20787274 | 4TU.ResearchData, 10.4121/20787274 |

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
