## [Editor Report]

This paper is of particular interest to researchers who plan to use focused-ion beam scanning electron microscopes (FIB-SEMs) and require fluorescent data to guide the milling process. The authors describe a valuable after-market upgrade that allows fluorescent data acquisition during FIB-milling without stage repositioning. Technical details of the fluorescent module upgrade together with the sample stage redesign are compellingly documented and will enhance the implementation of this important technology.

---

## [Decision Letter]

**Decision letter after peer review:**

Thank you for submitting your article "A cryogenic, coincident fluorescence, electron and ion beam microscope" for consideration by *eLife*. Your article has been reviewed by 2 peer reviewers, and the evaluation has been overseen by Suzanne Pfeffer as the Senior Editor. The following individual involved in the review of your submission has agreed to reveal their identity: Misha Kopylov (Reviewer #2).

Essential revisions:

Please respond to each of the reviewer comments by the inclusion of the responses within a textually revised manuscript.

*Reviewer #1 (Recommendations for the authors):*

I would have liked to see some self-reflection on the pros and cons of the internal FLM v.s and external FLM; e.g. one can use a very high-end confocal or super-resolution FLM externally and some comment on when this might be an advantage despite the risk of contamination would be helpful. Similarly, it would be nice to see some comments on how robust this system is. How long to set up the coincidence? How stable is this over time? How often do the authors think that a target will be hit rather than missed with full automation given a z-axis error of 90 nm? This is not to criticize the overall concept but just to try and keep the wider community from getting over-optimistic about the likelihood of full automation for these methods given the current limitations in accuracy.

I know how hard it is to get new instruments working and demonstrate viability so I would not ask for more biological results but am hoping that by the time the authors send the paper back with revisions they will be able to point to a few more successful biological cases.

I think that the word "solve" in this sentence is an over-exaggeration. "Here, we present in-situ fluorescence microscopy-guided FIB fabrication of a frozen-hydrated lamella to solve this problem"

In my opinion, the in situ cryoET method has over-promised results and underestimated challenges for a long time now and I would encourage the authors not to perpetuate this by claiming to have solved – rather than "addressed", "worked on reducing" etc. a set of problems. There are similar examples of mild over-statement elsewhere in the paper as well. The work is good enough not to need that to be convincing.

Are two hours to reach a stable cold temperature a long time? How does this compare to existing cold stages?

*Reviewer #2 (Recommendations for the authors):*

The paper is written in a very clear and precise style. Figures are very well put together and a good visual guide perfectly complements the main text.

Specific comments (questions) – text.

Line 99 – is this system only compatible with AutoGrids? Can slice-and-view, liftout from HPF hats, and waffle method be performed with this system?

Line 108 – it would be useful to have a shortlist of instruments from TFS and other manufacturers that are compatible or can be made compatible with the hardware.

Line 110 and 237 – could you comment on whether Odemis software allows automated lamellae preparation.

Line 120 – if required, can FLM objective be changed by the end-user or FSE to a lower magnification (5x, 20x, 63x, etc?).

Line 155 – can the "cold" time be extended beyond 9 hours using aftermarket solutions such as Delmic Cers Ice Shield?

Line 166 – what are the stage limits in relation to the FIB source?

Line 230 – can FLM images be acquired during the milling, without stopping the FIB?

Specific comments – figures.

Figure 1(a) – caption says: "EM hardware", perhaps "FIB-SEM hardware" will be better.

Figure 1(b) – this would be a good place to add stage limits at the coincidence point to answer the question "can I mill my sample at 5,10,15, 25 degrees" if necessary. Of particular interest to some groups may be the ability to orient the surface of the sample 90 degrees to FIB-beam to mill "precuts" following the Waffle method.

Figure 3(d) – it would be useful to see a zoomed-in graph around the 100 K to see minor temperature fluctuations.

Figure 3(f) – caption says: "The peak at 50 Hz originates from electrical interference", can you comment on the peaks around 500 Hz?

Figure 4(d) – the peak at around 100 Hz is even taller than 50 Hz, moreover, the peak is absent in the dummy stage (Figure 3f). Could you comment on this?

Figure 5(c) – caption says: "Adjustments to the fabrication procedure are easily made when making a lamella based on predefined milling patterns of i. e. 2, 1, 0.6, and 0.2 µm". This is unclear and seems to be unrelated to the actual figure. The figure seems to be showing milling progression as observed by FLM/RLM starting with intact (pre-mill) and going to progressively thinner lamellae.

Figure 6 – caption says: Scale bars 40, 10, 10, 5, 5 and 0.25 µm for respectively (a), (b), (d) left, (d) right, (e) and (f). This makes finding scale bars for panels unnecessarily complicated. Could you describe the scale bars one at a time such as: Scale bars: (a) – 40 µm, (b) – 10 µm, (d) left – 10 µm, (d) right – 5 µm, (e) – 5 µm, (f) – 0.25 µm. Also, for completeness, you could add scale bars in (c) and (g).

---

## [Author Response]

Reviewer #1 (Recommendations for the authors):I would have liked to see some self-reflection on the pros and cons of the internal FLM v.s and external FLM; e.g. one can use a very high-end confocal or super-resolution FLM externally and some comment on when this might be an advantage despite the risk of contamination would be helpful.

We thank Reviewer #1 for taking the time to review our manuscript, the kind comments and for appreciating the added value of our described system to the cryo-lamella preparation workflow.

We have added a Discussion section prior to the Conclusion, adding the following paragraphs, line 264:

“Currently, the system presented is solely compatible with samples in an AutoGrid form factor. With minor modifications, other planar sample types, such as samples prepared by the Waffle Method Kelley et al. (2022), would fit inside the sample shuttle. Identifying grid bars and sample navigation can be done directly using the LM. Imaging through a relatively thick sample (∼50 μm) pose potential challenges, as the increased imaging depth is likely to induce aberrations and increased scattering.

Although the system presented reduces the sample handling steps, contamination and avoids the use of fiducial markers, it also has its limitations: (i) Without an additional external cryo fluorescence microscope, FM can only be done when the sample is clipped and loaded into the FIB-SEM; (ii) A required working distance of ∼1 mm prevents high (>1) NA objective lenses to be used; (iii) Implementing a cryo-immersion objective lens adds significant complications and is not straight forward; (iv) The integrated approach reduces the number of available methods for super resolution microscopy due to the need for a compact optical path.

External cryogenic super resolution microscopy can readily provide sub diffraction limit imaging and can provide a milling target, but the localization precision will only be as good as the correlation accuracy provided by the fiducial markers images in both LM and FIB. This same limitation also applies to the different commercially available integrated LM systems that can be retrofitted onto the FIB-SEM, but require in-vacuum stage movements ThermoFisher Scientific (2021); Smeets et al. (2021).”

Similarly, it would be nice to see some comments on how robust this system is. How long to set up the coincidence?

This is mentioned in the ‘Material and Methods’ section (line 438):

“This process takes about 5 to 10min, and after this sample navigation and focusing is done through the sample stage.”

How stable is this over time? How often do the authors think that a target will be hit rather than missed with full automation given a z-axis error of 90 nm? This is not to criticize the overall concept but just to try and keep the wider community from getting over-optimistic about the likelihood of full automation for these methods given the current limitations in accuracy.

We have made textual changes at line 208:

“We found that the stability of the coincidence alignment was approximately 0.7, −0.2 and −0.05 μm / h for the *X*, *Y* and *Z* directions respectively. The largest drift is observed along the lamella width (*X*) which is typically not the most critical dimension, as the width usually spans several micrometers. See the Material and Methods section for more details.”

Supported by an additional figure (Figure 8) in the Material and Methods section and made textual changes at line 440:

“To test the stability of the coincidence alignment, we computed the cumulative image shift for all FIB-SEM images acquired for the stage repositioning measurements. We obtain the sample shift along the *X*, *Y* and *Z* directions as explained above. While the sample positioning stage performs random moves, it is aligned to the focal point of the optical objective when FIB-SEM images are acquired. The objective stage position is not actively moved or controlled. Hence, any objective stage drift, would result in *XY* shifts in the FIB-SEM images acquired. This is shown in Figure 8 as a function of time. The gaps in the recording are due to a lunch and a coffee break. At 1.7 h a new location on the sample was chosen to avoid bleaching of the fluorescent target bead. Along the *X* direction the drift was approximately ∼0.7 μm∕h and for the *Y* and *Z* direction this amounts to −0.2 and −0.05 μm∕h respectively.”

We have made textual changes in the discussion (line 287) to to provide an outlook:

“This would also open up the possibility for automated milling, based on acquired LM (astigmatic) data. For this to work, we would need to automatically correct for deviations in the coincident alignment or update the alignment regularly, but this requires more work to be done.”

I know how hard it is to get new instruments working and demonstrate viability so I would not ask for more biological results but am hoping that by the time the authors send the paper back with revisions they will be able to point to a few more successful biological cases.

The validation experiments have been performed at the MPI in Dortmund in 2021. At the time we had 4-5 different biological sample types shipped to Dortmund from labs to which the different co-authors belong. Some sample batches were affected by the international transfer and handling others are the same cell type, but only differ in labelling. We have diverse datasets for single types, as we loaded 3 or 4 AutoGrids for each (type). Since those experiments, we have been focusing our attention on doing cryo-ET ourselves in Delft but were (and still are) delayed by delivery of components and unfortunately we cannot deliver more data at this time. We have seen that the method is applicable to different cell types, and can be used with different fluorescent markers during the experiments, but do not have enough data to back this up in scientific literature.

I think that the word "solve" in this sentence is an over-exaggeration. "Here, we present in-situ fluorescence microscopy-guided FIB fabrication of a frozen-hydrated lamella to solve this problem"In my opinion, the in situ cryoET method has over-promised results and underestimated challenges for a long time now and I would encourage the authors not to perpetuate this by claiming to have solved – rather than "addressed", "worked on reducing" etc. a set of problems. There are similar examples of mild over-statement elsewhere in the paper as well. The work is good enough not to need that to be convincing.

That is a valid point, we made textual changes to the manuscript at line 25:

“… fabrication of a frozen-hydrated lamella to address this problem…”

and line 77:

“… thereby facilitating the application of cryo-CLEM during the cryo-ET workflow …”

Are two hours to reach a stable cold temperature a long time? How does this compare to existing cold stages?

To our knowledge, there are systems which cool down faster (i.e. cool-down of the cold stage of the TFS Aquilos typically takes about an hour). It very much depends on the thermal mass, type of cooling used, thermal links etc. The cryogenic chamber shield mentioned in the manuscript also takes about an hour to reach base temperature. Yes, 2 hours is longer than required on other systems, but we think it is not limiting provided the cooldown happens parallel to other preparations for the experiment.

Reviewer #2 (Recommendations for the authors):The paper is written in a very clear and precise style. Figures are very well put together and a good visual guide perfectly complements the main text.

We thank Reviewer #2 for taking the time to review our manuscript, the kind words regarding its style and the detailed comments.

Specific comments (questions) – text.Line 99 – is this system only compatible with AutoGrids? Can slice-and-view, liftout from HPF hats, and waffle method be performed with this system?

We have made textual changes to the manuscript at line 265:

“Currently, the system presented is solely compatible with samples in an AutoGrid form factor. With minor modifications, other planar sample types, such as samples prepared by the Waffle Method Kelley et al. (2022), would fit inside the sample shuttle. Identifying grid bars and sample navigation can be done directly using the LM. Imaging through a relatively thick sample (∼50 μm) pose potential challenges, as the increased imaging depth is likely to induce aberrations and increased scattering.”

Line 108 – it would be useful to have a shortlist of instruments from TFS and other manufacturers that are compatible or can be made compatible with the hardware.

We have made textual changes to the manuscript at line 106:

“The optical microscope hardware resides on a high vacuum (HV) door replacing the original door of the microscope, and it current design fits on a Thermo Fisher Scientific (TFS) SDB chamber. By adapting the design of the door, it will be compatible with dual beam systems retaining 52 degrees between SEM (top) and FIB (side) from other manufacturers.”

Line 110 and 237 – could you comment on whether Odemis software allows automated lamellae preparation.

Currently, Odemis does not allow for automated lamellae preparation, but the software is open source and available on GitHub at: https://github.com/delmic/odemis

Odemis is written in Python and offers control over microscopes from different manufactures (currently Zeiss, TFS, Tescan, Hitachi, JEOL IT-XXX). The level of remote control does depend on the exact API provided by the manufacturer. Though Odemis currently does not offer automation out-of-the-box, the hardware (LM, stages, etc) can be controlled solely through the Python back-end, which would allow for automation by combining this with i.e. SerialFIB or similar software.

Line 120 – if required, can FLM objective be changed by the end-user or FSE to a lower magnification (5x, 20x, 63x, etc?).

Objective lenses can be swapped provided that a parfocal length of 60 mm is retained, the minimal working distance is 1 millimeter. An adjustable correction collar is highly recommended for high-NA objectives.

Line 155 – can the "cold" time be extended beyond 9 hours using aftermarket solutions such as Delmic Cers Ice Shield?

A cryogenic chamber shield was used in these experiments (very similar to the shield mentioned). Slight changes to the microcooler design have since resulted in improved standing (‘cold’) times of ~15 hours, but have not been demonstrated in the full workflow, yet. In effect, the time it stays cold is limited by cryopumping water from the residual gas onto the cold surface of the microcooler. Less residual water in the rest gas (due to a big chamber shield) means longer standing time.

Line 166 – what are the stage limits in relation to the FIB source?

The angle of incidence of the FIB to the sample surface can be set between 9 and 40 degrees. The current sample holder design (hole in the side of the holder to allow FIB access) limits this to about 8-12 degrees. Note that the angle of the OL is not adjustable and is set to work at 10 degrees angle of incidence between FIB and sample surface.

Line 230 – can FLM images be acquired during the milling, without stopping the FIB?

Yes, this is shown in Video 1, which consists of three different recordings (a separate recording for each milling step). We stopped the recording after each step to facilitate image data organization. There is no principal reason one would need to stop the live feed from the LM during milling. The objective lens is protected against charged particle beam damage (either electrons or ions) by the glass coverslip below the sample (see figure 1b).

Specific comments – figures.Figure 1(a) – caption says: "EM hardware", perhaps "FIB-SEM hardware" will be better.

We have made textual changes to the manuscript at Figure 1(a) – caption:

“… separated from the FIB-SEM hardware in respectively the lower and upper half-spaces…”

Figure 1(b) – this would be a good place to add stage limits at the coincidence point to answer the question "can I mill my sample at 5,10,15, 25 degrees" if necessary. Of particular interest to some groups may be the ability to orient the surface of the sample 90 degrees to FIB-beam to mill "precuts" following the Waffle method.

We have made changes to the manuscript at Figure 1(b).

Figure 1(b) now shows settable incident angle range for FIB and SEM.

We have made textual changes to the manuscript at Figure 1(b) – caption:

“… The coincident point is formed by the electron-, ion- and light beams. Incident angles indicated are limited by the FIB access hole in the sample shuttle holder.”

We have made textual changes to the manuscript at line 170, adding the following sentence:

“It is designed to have ±3 mm translation range in *X*, *Y* , *Z* and ±28° rotation range in *Rx* when aligned to the FIB-SEM coincident point.”

Figure 3(d) – it would be useful to see a zoomed-in graph around the 100 K to see minor temperature fluctuations.

We have made changes to the manuscript at Figure 3(d).

An additional subplot now shows the temperature stability at 107 K. The ADC resolution of the Lakeshore Model 355 is 10 mK as shown by the few spikes deviating from 107.00 K. As cryo pumping water from the rest gas slowly increases the radiative heat load, we changed the set point to 106 K (not visible on this scale) halfway through the experiments to have little more heater power available for temperature stabilization.

Figure 3(f) – caption says: "The peak at 50 Hz originates from electrical interference", can you comment on the peaks around 500 Hz?

We have made textual changes to the manuscript line 163:

“Peaks around 500 Hz are caused by high pressure gas undergoing expansion through a restriction in the cold stage. An internal braid decouples these vibrations best along the *X* direction, but is stiffer along *Y* and *Z* directions.”

Figure 4(d) – the peak at around 100 Hz is even taller than 50 Hz, moreover, the peak is absent in the dummy stage (Figure 3f). Could you comment on this?

As we record images SEM images with varying line times to measure the mechanical vibration levels, it is possible that sampling artefacts occur. There are two contributions visible at 100 Hz, one exactly at 100 Hz (most likely electrical interference / sampling artefact) but also one peak at slightly higher frequency. Although the sample stage has been designed with high stiffness in mind, we hypothesize that the peak at ~110 Hz likely originates from reduced stiffness of the sample stage. This peak is absent if we use a dummy stage consisting of two aluminum blocks stacked on top of each other and bolted down to the optical module cassette deck (i.e. significantly stiffer than the actual piezo positioning stage).

Figure 5(c) – caption says: "Adjustments to the fabrication procedure are easily made when making a lamella based on predefined milling patterns of i. e. 2, 1, 0.6, and 0.2 µm". This is unclear and seems to be unrelated to the actual figure. The figure seems to be showing milling progression as observed by FLM/RLM starting with intact (pre-mill) and going to progressively thinner lamellae.

We have made textual changes to the manuscript at Figure 5(c) – caption.

“Stepwise milling progression as observed by RLM and FM. Estimated lamella thickness based on FIB images indicated at the top of the images. The fluorescent feature of interest is marked by the white cross and any misalignment (white arrows) can be corrected for.”

Figure 6 – caption says: Scale bars 40, 10, 10, 5, 5 and 0.25 µm for respectively (a), (b), (d) left, (d) right, (e) and (f). This makes finding scale bars for panels unnecessarily complicated. Could you describe the scale bars one at a time such as: Scale bars: (a) – 40 µm, (b) – 10 µm, (d) left – 10 µm, (d) right – 5 µm, (e) – 5 µm, (f) – 0.25 µm. Also, for completeness, you could add scale bars in (c) and (g).

We have made changes to the manuscript at Figure 3(d).

Scale bars have been added for subfigures (c) and (g).

We have made textual changes to the manuscript at Figure 6 – caption mentioning scale bars:

“Scale bars: (a) – 40 μm, (b) – 10 μm, (c) – 5 μm, (d) left – 10 μm, (d) right – 5 μm, (e) – 5 μm, (f) – 0.25 μm, (g) – 0.1 μm.”